# The Impact of Care Bundles on Ventilator-Associated Pneumonia (VAP) Prevention in Adult ICUs: A Systematic Review

**DOI:** 10.3390/antibiotics12020227

**Published:** 2023-01-20

**Authors:** Maria Mastrogianni, Theodoros Katsoulas, Petros Galanis, Anna Korompeli, Pavlos Myrianthefs

**Affiliations:** 1Department of Nursing, National and Kapodistrian University of Athens, 123 Papadiamantopoulou Str, Goudi, 11527 Athens, Greece; 2Department of Health Policy, Ministry of National Defense, Mesogeion Avenue 227–231, Holargos, 15561 Athens, Greece; 3Intensive Care Unit, General and Oncologic Hospital of Kifissia “AgioiAnargiri”, Kaliftaki 41 Str, Kifissia, 14564 Athens, Greece

**Keywords:** ventilator-associated pneumonia, care bundles, intensive care units, prevention

## Abstract

Ventilator-associated pneumonia (VAP) remains a common risk in mechanically ventilated patients. Different care bundles have been proposed to succeed VAP reduction. We aimed to identify the combined interventions that have been used to by ICUs worldwide from the implementation of “Institute for Healthcare Improvement Ventilator Bundle”, i.e., from December 2004. A search was performed on the PubMed, Scopus and Science Direct databases. Finally, 38 studies met our inclusion criteria. The most common interventions monitored in the care bundles were sedation and weaning protocols, semi-recumbent positioning, oral and hand hygiene, peptic ulcer disease and deep venus thrombosis prophylaxis, subglottic suctioning, and cuff pressure control. Head-of-bed elevation was implemented by almost all studies, followed by oral hygiene, which was the second extensively used intervention. Four studies indicated a low VAP reduction, while 22 studies found an over 36% VAP decline, and in ten of them, the decrease was over 65%. Four of these studies indicated zero or nearly zero after intervention VAP rates. The studies with the highest VAP reduction adopted the “IHI Ventilator Bundle” combined with adequate endotracheal tube cuff pressure and subglottic suctioning. Multifaced techniques can lead to VAP reduction at a great extent. Multidisciplinary measures combined with long-lasting education programs and measurement of bundle’s compliance should be the gold standard combination.

## 1. Introduction

Ventilator-associated pneumonia (VAP) is one of the main types of infection in critically ill mechanically ventilated patients, leading to increased mortality, morbidity, hospital stay, economic and psychological costs for patients and their families [1,2,3,4]. During the last two decades, many guidelines have been proposed to reduce the incidence of VAP. It has been scientifically proven that interventions must be combined in order to be useful [5,6,7]. Bundle is a set of individual components, combined to make a set of quality indicators for a specific system, procedure, or treatment [8]. These interventions must be all implemented together to achieve significantly better results [9].

In 1983, the CDC published the guidelines for the prevention of nosocomial pneumonia, which were specialized for VAP in 2003 [9]. In December 2004, the Institute for Healthcare Improvement (IHI), during the promotion of the “100,000 Lives Campaign”, inserted the “IHI Ventilator Bundle”, consisting of four elements: (1) elevation of the head of bed (HOB) to 30°–45°; (2) daily “sedation vacation” and assessment of readiness to extubate; (3) peptic ulcer disease (PUD) prophylaxis and (4) deep venus thrombosis (DVT) prophylaxis [9]. In 2010, IHI added a fifth intervention: (5) daily oral care with chlorhexidine. In 2016, the Intensive Care Society proposed a bundle called “Recommended bundle of Interventions for the prevention of VAP”, including elevation of head of bed, daily sedation vacation and assessment of readiness to extubate, use of subglottic secretion drainage, avoidance of scheduled ventilator circuit changes, oral hygiene without chlorhexidine and PUD prophylaxis (only for high-risk patients), without mentioning DVT prophylaxis. Next year, the European Respiratory Society, in collaboration with the European Society of Intensive Care Medicine, the European Society of Clinical Microbiology and Infectious Diseases and the American Latin Thoracic Association published the “International guidelines for the management of VAP”, introducing the use of Selective Digestive and Oropharyngeal Decontamination and proposing Oral Decontamination without chlorhexidine. Most of the ICUs worldwide adopted the “IHI Ventilator Bundle”, adapting it to their own needs. As a result, there has been a variation in the included interventions of VAP bundles among ICUs and until now there is no common bundle which can be agreed to be implemented by the communities worldwide [10].

In the last two decades, a major problem has been the increasing rates of occurrence of community-associated methicillin resistant *Staphylococcus aureus* (CA-MRSA) and hospital-associated MRSA (HA-MRSA). Apart from that, the communities have faced several viral outbreaks, such as SARS-1, SARS-2, and MERS. All the above severe respiratory syndromes and population aging have led to increased rates of ventilated patients [11]. As VAP is one of the most common preventable lung infections in critically ill intubated patients, it is imperative to determine the most efficient preventive measures for VAP reduction. When there is such heterogeneity in ventilator bundles, the question is which ventilator bundle may be more effective?

A great number of studies examined the effectiveness of the different combinations of interventions for VAP prevention. Several reviews have summarized the findings of those studies [11,12,13,14]. A previous systematic review [11] synthesized all the VAP bundles; however, it was published seven years ago, and several studies have been published since then. Moreover, the most recent review was published in 2022, but its main purpose was to summarize the strategies for improvement of care bundle compliance [14]. Therefore, the aim of our systematic review was to present all the multidimensional interventions used in ICUs, from the introduction of the “IHI Ventilator Bundle”, i.e., from December 2004, which is chronologically a point of intersection, as most of the ICUs started to adopt the “IHI Ventilator Bundle”, either alone or combined with other interventions in order to prevent VAP.

## 2. Materials and Methods

### 2.1. Data Sources and Strategies

A systematic literature review was conducted in accordance with the Preferred Reporting Items for Systematic Reviews and Meta-Analysis (PRISMA) guidelines [15]. A comprehensive search was performed on the PubMed, Scopus, and Science Direct databases. We used the following strategy: “ventilator-associated pneumonia” AND “care bundles” AND ”intensive care units” AND ”prevention”. Databases were searched from January 2005 to 23 October 2022. The selected period was intended because in December 2004, the IHI introduced the “IHI Ventilator Bundle”. This systematic review has been registered on PROSPERO registry (ID: CRD42022384828).

### 2.2. Selection and Eligibility Criteria

Three independent authors were responsible for removing the duplicates, screening the title and abstract, and analyzing the full content of the studies in accordance with the inclusion criteria. Two independent authors selected the included studies and another independent author resolved possible disagreements. To be included in our systematic review, studies had to (a) be published in the English language; (b) be pre–post observational studies; (c) include adult critically ill patients, intubated at least 48 h, admitted to all kinds of adult ICUs; (d) evaluate the implementation of care bundles in VAP prevention by thoroughly presenting all the combined interventions and calculating the pre and post intervention VAP rate; (e) compare with the individual intervention’s implementation for VAP prevention; and (f) be published after the implementation of “IHI Ventilator Bundle”. Additionally, we excluded protocols, conference papers, abstracts, posters, and letters to editors and editorials.

### 2.3. Data Extraction and Risk of Bias Assessment

For each study, the main characteristics were extracted: authors, country, data collection period, study setting, sample size, age of patients, measures, implementation of an educational program, and main findings. The appraisal of quality of the included studies was performed by using the Risk of Bias in Non-randomized Studies of Interventions (ROBINS-I) [16]. The checklist consisted of seven domains of bias: confounding, participants’ selection, interventions’ classification, deviation from intended interventions, missing data, outcomes’ measurement, and reporting biases. After completing the checklist, all studies were classified as low risk, moderate risk, serious risk, critical risk, or “no information”.

## 3. Results

### 3.1. Identification and Selection of Studies

The study selection process is demonstrated in Figure 1. After an initial database search, a total of 3984 studies were identified. After title, abstract, and full content screening, 38 studies were finally included in the review.

### 3.2. Characteristics of the Studies Included in This Review

Finally, 38 studies were included in our review. The main characteristics of the studies are presented in Table 1. Fifteen studies took place in Asia, ten in Europe, six in South America, five in North America, one in Australia, and one in Africa. The data collection period ranged from October 2003 [17] to February 2021 [18]. Two studies [5,19] were conducted in Saudi Arabia, at the period of MERS outburst and one study [18] was carried out in Egypt during the coronavirus pandemic. The total sample size ranged from 43 intubated patients [20] to 171,237 intubated patients [21]. All the studies, due to the inclusion criteria, had a pre- and post-intervention observational study design. The studied populations were critically ill ventilated patients admitted to general, medical, surgical, neurosurgical, trauma, and cardiovascular ICUs. Most of the included studies were performed in general ICUs and fifteen of them were multicenter.

### 3.3. Risk of Bias Assessment

The risk of bias assessment of the reviewed studies is demonstrated in Appendix A. The risk of bias was low in seven studies [2,5,22,23,24,25,26], serious and critical in 11 studies [3,6,7,17,19,27,28,29,30,31] and moderate in the remaining studies. The most common bias in the pre-post observational studies was the bias due to deviations from intended interventions.

### 3.4. Ventilator-Associated Care Bundles—Patients’ Outcomes

All of the studies used multifaceted strategies to prevent VAP, as it was one of the inclusion criteria. The number of the included ventilator bundle elements varied from four to thirteen and is shown in Table 2. Head-of-Bed elevation, with a range of 30° to 45°, was implemented by all the reviewed studies, except one [32]. The second most widely used intervention was oral hygiene using chlorhexidine 0.12%. Only one study [3] used sodium bicarbonate and another one [33] sponges and mouthwashes, albeit without a particular change in VAP reduction. Six studies did not adopt the measure of oral care [3,17,18,29,30,32]. Nine studies adopted the IHI Ventilator Bundle [5,6,19,20,23,33,34,35,36]. In two studies, one additional preventive measure was selective oropharyngeal decontamination by using colistin, tobramycin, and nystatin, three times daily, with remarkable VAP reduction rates [25,37].

Patients’ outcomes are shown in Table 3. Thirty-six of the included studies indicated a reduction in VAP incidence. VAP reduction rates ranged from 13% [2] to 100% [23,36]. Only four studies showed low VAP reduction [2,38,39,40]. The majority of the studies showed a reduction of 36–64% and twelve of them a reduction over 65% [5,6,19,20,23,24,25,27,29,30,36,41]. None of the studies that found low rates of VAP reduction, used the “IHI Ventilator Bundle” and more specifically, all of them did not adopt the PUD and DVT prophylaxis, with three of them adopting the rest measures of the “IHI Ventilator Bundle” (daily sedation vacation, daily assessment of readiness for extubation, head-of-bed elevation and oral care with chlorhexidine). Fourteen studies [2,5,17,18,19,20,21,22,25,35,38,40,42,43] included subglottic suctioning in their bundles, with four of them [5,19,20,25] to achieve more than 65% VAP reduction.

In nine studies [7,17,19,21,23,24,26,30,42], statistical significance was not mentioned, and in four studies [2,32,38,43], the results were not statistically significant, while 24 studies reported a statistically significant reduction. One single center study [43] showed that there were no significant differences in after-intervention VAP and early onset VAP rates but only in late onset VAP rates. In our systematic review, *p*-values of less than 0.05 were considered significant.

The studies with the highest rate of reduction [5,19,25] implemented the “IHI Ventilator Bundle” combined with adequate ETT cuff pressure at 20–30 cm H_2_O and subglottic suctioning. Moreover, the study of Gallagher et al. (2012) [36] indicated zero after intervention VAP rate, giving great importance not only to the adoption of “IHI Ventilator Bundle” but also to hand washing and condensate removal. Another study with zero after intervention VAP rate was the study of Chen et al. (2014) [23], where “IHI Ventilator Bundle” with adequate ETT cuff pressure were used as preventive measures.

### 3.5. Educational Program

We identified thirty-three studies that adopted an education program, as demonstrated in Appendix A. Education program included self-learning packets, presentations, discussions, knowledge questionnaires, posters, checklists, videos, reminder signs in patients’ rooms, nursing and medical champions, stimulation scenarios and feedback meetings. Thirty-two studies measured the compliance to ventilator bundles and/or education program, and in all of them there was a significant increase in VAP bundle adherence after the education program.

## 4. Discussion

Our systematic review demonstrated the variety of interventions included in ventilator care bundles and their implementation in VAP decrease in adult ICUs. Thirty-eight studies met our inclusion criteria and we found that the combined measures can prevent VAP toa greater extent. Moreover, our review showed that the number of intervention strategies varied widely in the included studies. Most of the studies examined whether ventilator bundle’s compliance could be increased through health workers’ training.

VAP is one of the main healthcare-associated infections in intubated critically ill patients. VAP is associated with an exceeded duration of mechanical ventilation. The most common microorganisms responsible for VAP are pseudomonas aeruginosa, escherichiacoli, klebsiella pneumoniae and the Acinetobacter species from Gram-negative microorganisms and staphylococcus aureus from Gram-positive microorganisms. VAP diagnostic criteria differ among ICUs but usually require factors such as fever, leukocytosis, progressive infiltrate on chest X-ray, positive cultures from respiratory secretions and reduction in gas exchange [1,9].

During the last decades, there has been a great scientific concern about finding the best strategies to prevent VAP and ventilator-associated events [1]. Worldwide, scientists’ aim to decrease VAP incidence in order: first, to improve ventilated patients’ outcomes, and thereafter to decrease mortality, hospital length of stay and healthcare expenditures. VAP diagnostic criteria, methods of respiratory sampling, interventions in VAP bundles and study designs varies, at a great extent, among the ICUs worldwide [24]. Nevertheless, VAP care bundles combined with a focused education program, which could lead to increased compliance, have proved their efficacy in VAP reduction [6].

A multimodal approach for VAP prevention includes functional, mechanical, and pharmacological measures [21]. The most important intervention for preventing VAP, is avoiding intubation, by using noninvasive positive pressure ventilation whenever possible [1]. By the review of the existing studies, the most frequent proposed interventions, when the patient should be intubated, were sedation and weaning protocols, semi-recumbent positioning, oral and hand hygiene, PUD and DVT prophylaxis, subglottic suctioning, and cuff pressure control. However, interventions as, avoidance of nasogastric tubes, nasotracheal intubation, accidental extubation and gastric overdistension, aseptic suctioning technique and adherence to recommended frequency of equipment change were monitored in the minority of the included studies. Two studies used as an additional preventive measure Selective Oropharyngeal Decontamination (SOD) [25,37]. The care bundle without SOD was associated with a decrease of 42% in VAP rates, while with the implementation of SOD there was a further decrease of 70% in VAP rates [25].

A semi-recumbent position was established with the introduction of “IHI Ventilator Bundle”, though it was proposed much earlier in studies conducted from 1992 to 1999 to test its contribution in aspiration prevention [9]. The implementation of this measure by almost all studies shows the acceptance and the recognition of its importance in VAP treatment. In our systematic review, most of the studies involved oral hygiene in their bundles. Oral care remains an important tool for dental plaque removal and the promotion of a normal microbial community inside the oral cavity, thus preventing the growth of microorganisms in the trachea and the creation of VAP [44,45,46,47]. Fourteen studies aimed to figure out the benefits of subglottic secretion drainage as the secretions concentrated between the vocal cords and the ETT may cause VAP because of the potential growth of pathogens. These studies showed that subglottic suctioning seem to be a useful preventive measure.

A notable result of our study is that ten studies did not control the daily readiness for extubating, when the presence of the endotracheal tube is one of the major predisposing factors for VAP development, as pathogens enter the trachea either by micro aspiration around the ETT cuff or by the biofilm formed in the inner side of the ETT. Moreover, only nine of the included studies followed the “IHI VAP preventive Bundle”. All of them, achieved a VAP reduction at least 36%, five of them showed a reduction of over 65% and two of them managed a zero after intervention VAP rate. These results may indicate the important contribution of the IHI VAP preventive Bundle, alone or in combination with other measures, in VAP reduction. It is noteworthy that the revised studies, carried out the last six years, did not use the “IHI Ventilator Bundle” but adopted some of its interventions.

Another thing that was remarkable was that in many studies, hand hygiene was not mentioned to be monitored as an element in the VAP care bundles. We assume that measures such as hand hygiene and aseptic suctioning technique are of the most basic techniques and that they were taken for granted, along with all the other interventions for VAP prevention in the included studies. Another interesting conclusion in some studies [19,23,28,36] was the after intervention zero-VAP rate, which could be due to many factors, such as country health care policy makers, diversity in VAP diagnostic criteria and methods of BAL sampling [48].

Over the past two decades, humanity has faced several viral outbreaks, causing severe respiratory syndromes, such as SARS-1, SARS-2 and MERS, which may have increased the percentage of intubated patients at a local level. According to the inclusion criteria, all the included studies were conducted after December 2004. Taking into consideration the data collection periods of the revised studies, in two studies [5,19] carried out the period of MERS outbreak in Saudi Arabia, the viral load of the study population and the patient outcomes might have been influenced. Furthermore, only one of the included studies [18] has been performed since the beginning of the coronavirus pandemic. In all three studies, the authors did not mention the possibility of any influence in patients’ outcomes. The data collection periods of the remaining studies did not coincide with the above viral outbreaks’ periods.

The study of Alvarez-Lerma et al. (2018) [21] was the largest study, including 181 ICUs and 171,237 patients, demonstrating the effectiveness of implementing a VAP prevention bundle at a national level. The study of Kao et al. (2019) [39] indicated that compliance rates of VAP bundle care and VAP reduction rates differ between the different types of ICUs (medical, surgical, cardiovascular). Three studies [41,49,50] adopted the INICC multidimensional approach, including the following practices: (1) bundle for VAP prevention (2) education (3) outcome surveillance (4) process surveillance (5) feedback on VAP rates and (6) performance feedback of infection control practices.

Finally, most of the included studies, apart from the implementation of care bundles, adopted at the same time a multifaced educational program. Presentations, posters, videos, discussions, stimulating scenarios and feedback meetings were some of the ways that ICU staff was trained. Our systematic review underlined the wide range in VAP reduction. This wide fluctuation may have been affected, to some extent, by either the country’s healthcare system or the extent variety of educational programs and different degree of personnel’s adherence, but at the same time the studies’ results highlighted that multidisciplinary health professionals’ education and frequent monitoring of adherence can lead to better patients’ outcomes.

## 5. Limitations

There are several limitations in this systematic review. At first, as we searched for relevant studies only in English language, other potential studies written in different languages were not included in our review. Also, we included three databases in our research, and thus additional studies could be found in other databases. In addition, application of vigorous inclusion and exclusion criteria may preclude some studies to be included in our review. Moreover, among the reviewed studies, there was a heterogeneity, to a greater extent, of the study settings. In particular, differences in study quality are indicative of great differences in studies design, since seven studies had a high level of quality, 20 had a moderate level, and 11 a low level. However, we should notice that heterogeneity among studies was not a limitation of our review but an unavoidable effect of the studies design. Therefore, we should consider our results with caution since it is difficult to establish solid conclusions.

## 6. Conclusions

From our systematic review, it emerged that there is a considerable variation between the VAP care bundles and education programs in ICUs worldwide. This variation leads to discrepancies and does not allow comparability, to a greater extent, between the studies to find the gold standard VAP care bundle. The IHI VAP preventive Bundle seems to be a very useful tool in VAP reduction, combined with adequate ETT cuff pressure and subglottic suctioning, without forgetting the hand hygiene and aseptic suctioning technique. ICUs should adopt basic practices that prevent or decrease VAP rates, and as a result, mortality, duration of mechanical ventilation, length of stay, and healthcare costs. Moreover, the strategies should be multifaceted and supported by a long-term education program by ensuring compliance in the care bundle. These multidisciplinary strategies and education programs should be common in all ICUs. At least in the same country, national or cross-sectional randomized controlled studies need to be carried out in order to be able to compare the measures’ efficacy and find the best combination of preventive interventions.

## Figures and Tables

**Figure 1 antibiotics-12-00227-f001:**
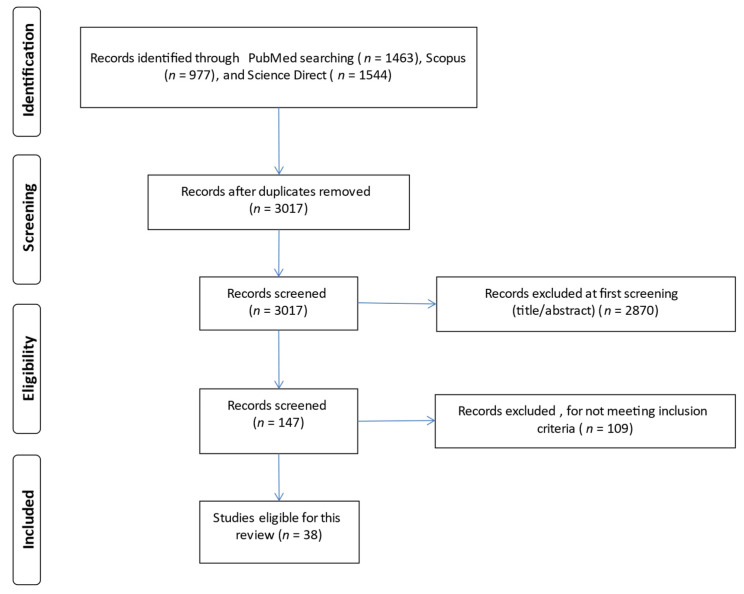
Flow diagram according to the Preferred Reporting Items for Systematic Reviews and Meta-Analysis.

**Table 1 antibiotics-12-00227-t001:** Main characteristics of the reviewed studies (n = 38).

ID	Reference	Country	Data Collection Period	Study Setting	Sample Size	Mean Age of Patients (Standard Deviation)	Educational Program (Yes/No/NA)
1.	Al-Tawfiq et al. (2010)	Saudi Arabia	1 January 2006–31 December 2008	One 18-bed ICU	NA	NA	yes
2.	Bouadma et al. (2010)	France	2-year period (Months and year NA)	One 20-bed MICU	1649 ventilator days	NA	yes
3.	Bird et al. (2010)	USA	1 March 2006–31 May 2009	Two SICUs	NA	NA	NA
4.	Ban et al. (2011)	Korea	31 October 2005–28 February 2006	One ICU	155 patients	NA	yes
5.	Berenholtz et al. (2011)	USA	1 October 2003–30 September 2005	81 ICUs	550,800 ventilator days	NA	yes
6.	Morris et al. (2011)	Scotland	NA	One 18-bed ICU	pre: 1460 patients post: 501 patients	pre: 60 (47–72) * post: 59 (48–70) *	yes
7.	Gallagher et al. (2012)	USA	31 August 2010–30 September 2011	One ICU	83 patients	63 (NA)	yes
8.	Moore et al. (2012)	USA	1 January 2011–30 June 2012	One 16-bed combined Neurosurgical & Trauma ICU	1987 patients	NA	NA
9.	Gatell et al. (2012)	Spain	1 January 2008–31 May 2009	One 16-bed GICU	NA	NA	yes
10.	Guanche-Garcell et al. (2013)	Cuba	31 January 2007–30 November 2010	One ICU	pre: 67 patients post: 1008 patients	pre: 60 (17.6) post: 61.4 (17.6)	yes
11.	Leblebicioglu et al. (2013)	Turkey	31 August 2003–31 January 2009	11 ICUs	pre: 448 patients post: 3864 patients	pre: 52.4 (22.5) post: 49 (21.6)	yes
12.	Mehta et al. (2013)	India	31 July 2004–31 October 2011	21 ICUs	pre: 3979 patients post: 42,966 patients	pre: 54.8 (17.8) post: 54.5 (18.3)	yes
13.	Micik et al. (2013)	Australia	1 April 2011–31 August 2012	One 8-bed cardiothoracic ICU	NA	NA	yes
14.	Viana et al. (2013)	Brazil	1 January 2014–30 June 2008	One 14-bed ICU	pre: 294 patients post: 224 patients	pre: 77 (65–85) * post: 76 (61–83) *	NA
15.	Chen et al. (2014)	Taiwan	1 January 2010–31 December 2012	One MICU & one SICU	NA	NA	yes
16.	Docher et al. (2014)	USA	1 April 2009–30 September 2012	One 18-bed MICU	713 patients	58.8 (17.5)	yes
17.	Eom et al. (2014)	Korea	31 July 2010–30 June 2011	6 ICUs	NA	NA	yes
18.	Righi et al. (2014)	Italy	31 January 2004–31 December 2010	One 10-bed ICU	1372 patients	61.1 (17.1)	NA
19.	Ismail et al. (2015)	Lebanon	NA	One CCU	pre: 15 patients post: 28 patients	pre: 67.1 (16.5) post: 56.2 (25.7)	yes
20.	Lim et al. (2015)	Taiwan	1 January 2006–31 March 2013	5 SICUs	27,125 patients pre: 12,913 patients post: 14,212 patients	pre: 63.2 (NA) post: 62.8 (NA)	yes
21.	Zeng et al. (2015)	China	1 December 2011–31 May 2014	One MICU	375 patients	NA	NA
22.	Alcan et al. (2016)	Turkey	7 April 2014–31 October 2014	One GICU	128 patients	NA	yes
23.	Khan et al. (2016)	Saudi Arabia	2008–2013	One GICU	3665 patients	53.2 (21)	yes
24.	Mogyorodi et al. (2016)	Hungary	1 January 2015–31 December 2015	One 12-bed ICU	535 patients pre: 275 patients post: 260 patients	pre: 69.8 (14.3) post: 68.7 (14.0)	yes
25.	Marini et al. (2016)	Saudi Arabia	31 October 2012–30 June 2014	One GICU	NA	NA	yes
26.	Parisi et al. (2016)	Greece	2–year study period	One 30-bed ICU	pre: 226 patients post: 136 patients	pre: 59 (41–73) * post: 58 (42–72) *	yes
27.	Alvarez–Lerma et al. (2018)	Spain	1 April 2011–31 December 2012	One hundred eighty-one ICUs	171,237 patients	NA	yes
28.	Burja et al. (2018)	Slovenia	1 September 2014–30 April 2015	One 12-bed MICU	pre: 55 patients post: 74 patients	pre: 67.8 (14.5) post: 64.8 (13.7)	yes
29.	Landelle et al. (2018)	Switzerland	31 August 2014–31 July 2016	One 34-bed ICU	pre: 291 patients post: 356 patients	pre: 61.9 (48.6–73.4) * post: 60.5 (49.4–71.2) *	yes
30.	Cengiz et al. (2019)	Turkey	1 January 2015–30 January 2016	9 ICUs	NA	NA	yes
31.	Kao et al. (2019)	Taiwan	1 January 2012–31 October 2014	7 SICUs, one CV/SICU, two MICUs	NA	NA	yes
32.	Sousa et al. (2019)	Portugal	31 October 2015–31 March 2017	3 ICUs	828 patients	NA	yes
33.	Branco et al. (2020)	Brazil	30 June 2017–30 June 2018	One GICU	302 patients	62.4 (17.1)	yes
34.	Fortaleza et al. (2020)	Brazil	1 January 2007–30 June 2019	Two ICUs	NA	NA	yes
35.	Liu et al. (2020)	China	1 June 2017–31 May 2019	6 ICUs	4716 patients	NA	NA
36.	Michelangelo et al. (2020)	Argentina	31 January 2016–31 December 2018	3 ICUs	NA	NA	yes
37.	Ochoa-Hein et al. (2020)	Mexico	2015–2018	One 14-bed ICU	NA	NA	yes
38.	Shaban et al. (2021)	Egypt	31 March 2020–28 February 2021	Two ICUs	pre: 52 patients post: 52 patients	pre: 58.4 (4.4) post: 57.8 (2.9)	NA

Abbreviations: CCU: Critical Care Unit; CV: Cardiovascular; GICU: General Intensive Care Unit; ICU: Intensive Care Unit; IP: Intervention Phase; MICU: Medical Intensive Care Unit; NA: Not Applicable; SICU: Surgical Intensive Care Unit. * median (interquartile range).

**Table 2 antibiotics-12-00227-t002:** Preventive VAP measures monitored in revised studies.

ID	Reference	Physicians’ Interventions						Nurses’ Interventions														
				IHI Ventilator				Bundle														
		Avoid Nasogastric Tube	Avoid Nasotracheal Intubation	PUD Prophylaxis	DVT Prophylaxis	Daily Sedation Vacation	Daily Assessment of Readiness for Extubating	HOB Elevation 30°–45°	Oral Care/Chlorhexidine 0.12%	Adequate ETT Cuff Pressure (20–30 cm H_2_O)	Subglottic Suctioning	Hand Hygiene	Aseptic Suctioning Technique	Avoid Accidental Extubation	Avoid Gastric Overdistension	Adherence to Recommended Frequency of Equipment Change **	Suction When Necessary	Suitable Use & Replacement of HME Filters	Closed Suction System	Change Soiled/Damaged VC	Condensate Removal	Other
1.	Al-Tawfiq et al. (2010)			√	√	√	√	√				√	√									
2.	Bird et al. (2010)			√	√	√	√	√														
3.	Bouadma et al. (2010)	√						√	√	√		√			√		√					gloves gowns
4.	Ban et al. (2011)											√	√			√	√	√		√		gloves
5.	Berenholtz et al. (2011)	√		√	√	√	√	√			√							√	√	√		
6.	Gallagher et al. (2012)			√	√	√	√	√	√			√									√	
7.	Morris et al. (2011)					√	√	√	√													
8.	Moore et al. (2012)			√	√	√		√	√													
9.	Gatell et al. (2012)							√	√	√	√	√	√	√	√							(b)
10.	Guanche-Garcell et al. (2013)		√	√		√	√	√	√	√		√			√				√	√	√	(c)
11.	Leblebicioglu et al. (2013)		√				√	√	√	√		√			√				√	√	√	(c) (j)
12.	Mehta et al. (2013)		√				√	√	√	√		√			√				√	√	√	(c) (j)
13.	Micik et al. (2013)					√	√	√	√	√	√	√		√	√	√	√		√			
14.	Viana et al. (2013)			√	√	√	√	√	√ ****													
15.	Chen et al. (2014)			√	√	√	√	√	√	√												
16.	Docher et al. (2014)			√	√	√	√	√	√		√											
17.	Eom et al. (2014)			√	√			√	√													
18.	Righi et al. (2014)		√				√	√	√	√		√			√	√	√		√			SDD
19.	Ismail et al. (2015)			√	√	√	√	√	√		√	√										gloves
20.	Lim et al. (2015)			√	√	√	√	√	√	√		√										(a)
21.	Zeng et al. (2015)					√	√	√	√	√		√				√	√					(l)
22.	Alcan et al. (2016)			√	√	√	√	√	√	√		√										
23.	Khan et al. (2016)			√	√	√	√	√	√	√	√											
24.	Mogyorodi et al. (2016)							√	√	√		√	√	√				√		√	√	
25.	Marini et al. (2016)			√	√	√	√	√	√	√	√											
26.	Parisi et al. (2016)			√	√	√	√	√		√ ***												
27.	Alvarez-Lerma et al. (2018)							√	√	√	√	√						√		√		(e)
28.	Burja et al. (2018)			√			√	√	√	√	√								√			
29.	Landelle et al. (2018)					√	√	√	√	√	√	√										(i) SOD
30.	Cengiz et al. (2019)					√	√	√	√	√	√	√			√			√	√			(f)
31.	Kao et al. (2019)					√	√	√	√	√		√										(h)
32.	Sousa et al. (2019)					√	√	√	√	√	√									√		(c) (i)
33.	Branco et al. (2020)							√	√	√								√		√	√	
34.	Fortaleza et al. (2020)					√		√	√	√												(h)
35.	Liu et al. (2020)					√	√	√	√	√	√	√	√									
36.	Michelangelo et al. (2020)						√	√	√	√												
37.	Ochoa-Hein et al. (2020)					√ *	√ *	√	√	√		√	√				√					(k)
38.	Shaban et al. (2021)							√		√	√	√								√		

Abbreviations: HOB: Head-of-bed; PUD: Peptic Ulcer Disease; DVT: Deep Vein Thrombosis; ETT: Endotracheal Tube; VC: Ventilator Circuit; SOD: Selective Oropharyngeal Decontamination (with colistin, tobramycin, nystatin, 3 times/day); IHI: Institute of Healthcare Improvement; SDD: Selective Digestive Track Decontamination; ** Adherence to recommended frequency of equipment change: HME filter 48 h, breathing circuit only solid, closed suction system 72 h; *** With sodium bicarbonate;**** With sponges & mouthwashes/no chlorhexidine; (a): high-level sterilization andstorage of the ventilator tubing; moisten the devices with sterile water; (b): smallest possible calibre nasogastric tube; (c): non-invasive positive pressure ventilator; (e): procedures andprotocols to reduce duration of MV; selective decontamination of the oropharyngeal and the digestive tract; (f): not using routine saline solution in aspiration; confirm feeding tube placement; (h): emptying water from the respirator tube; (i): active mobilization; (j): avoidance of histamine receptor 2 (H2)-blocking agents & proton pump inhibitors; use of sterile water to rinse reusable respiratory equipment; (k): tooth brushing; patient position changes; non-invasive ventilation in selected patients (acute cardiogenic pulmonary edema andtype 2 respiratory insufficiency in patients with chronic obstructive pulmonary disease (introduced in January 2016); * and use of non-invasive ventilatory assistance or high—flow nasal cannula in extubated patients (introduced in January 2018); (l): personal protective equipment for suctioning, daily cleaning of the ventilator andsuction bottle with sterile distilled water; sterilization of the circuit by pasteurization; use of an independent care room.

**Table 3 antibiotics-12-00227-t003:** Patient outcomes in each study.

ID	Reference	Pre-Intervention VAP	Post-Intervention VAP	*p*-Value	Comments
1.	Al-Tawfiq et al. (2010)	9.3	1-year after: 2.3 2-years after: 2.2	*p* < 0.001	
2.	Bouadma et al. (2010)	23.5 (26.7%)	1-year after: 14.9 (15.3%) 2-years after: 11.5(11.1%)	*p* < 0.0001	
3.	Bird et al. (2010)	10.2	3.4	NA	
4.	Ban et al. (2011)	17.4	11.04	*p* = 0.074	
5.	Berenholz et al. (2011)	6.9	16-months after: 3.4 28–30 months after: 2.4	NA	
6.	Morris et al. (2011)	32.0	12.0	*p* < 0.001	
7.	Gallagher et al. (2012)	25.5	0.0	*p* = 0.003	
8.	Moore et al. (2012)	4.5	Ranged per quarter	NA	The VAP rate per quarter (total 6 quarters) ranged from 1.94 to 6.55 (M = 4.33, SD: 1.65)
9.	Gatell et al. (2012)	9.9	9.3	*p* = 0.36	VAP incidence (>4 days after intubation): 4.6 vs. 3.1
10.	Guanche-Garcell et al. (2013)	52.6	15.3	*p* = 0.003	70% reduction
11.	Leblebicioglu et al. (2013)	31.1	16.8	*p* = 0.0001	
12.	Mehta et al. (2013)	17.4	10.8	*p* = 0.0001	38% reduction
13.	Micik et al. (2013)	13.4	7.7	NA	
14.	Viana et al. (2013)	18.6	11.8	*p* = 0.002	
15.	Chen et al. (2014)	1.5	0.0	NA	
16.	Docher et al. (2014)	9.3	Ranged per month	*p* < 0.001	●Mean after IVR: 3.2 (SD: 5.71). ●Average VAP reduction/month: 0.27
17.	Eom et al. (2014)	4.08	1.16	NA	
18.	Righi et al. (2014)	15.9%	6.7%	*p* < 0.001	* VAP bundle period: 2004–2007 * VAP bundle & SOD period: 2008–2010 ●EVAP (6.6% to 1.9%) ●LVAP (9.3% to 4.7%)
19.	Ismail et al. (2015)	66.7%	21.4%	*p* = 0.003	
20.	Lim et al. (2015)	13.63	3.9	*p* < 0.001	●Ventilator utilization ratio decreased by 9.9% & VAP density reduced by 1.9 cases/1000 ventilator days (up to a 57.6% reduction)
21.	Zeng et al. (2015)	0.495	0.281	*p* = 0.001	
22.	Alcan et al. (2016)	15.91	8.50	*p* = 0.0001	
23.	Khan et al. (2016)	8.6	2.0	*p* < 0.001	
24.	Mogyorodi et al. (2016)	21.5 (95% CI: 14.17–31.10)	12.0 (95% CI: 7.2–19.49)	NA	Relative risk reduction: 44% (95% CI: −0.5 to 0.98)
25.	Marini et al. (2016)	4.0	0.8	NA	
26.	Parisi et al. (2016)	21.6	11.6	*p* = 0.01	
27.	Alvarez-Lerma et al. (2018)	9.83 (95% CI: 8.42–11.48)	4.34 (95% CI: 3.22–5.84)	NA	
28.	Burja et al. (2018)	Total: 41.8% EVAP: 10.9% LVAP: 30.9%	Total: 25.7% EVAP: 12.2% LVAP: 13.5%	Total: *p* = 0.061 EVAP: *p* > 0.99 LVAP: 0.027	
29.	Landelle et al. (2018)	24.0	3.9	*p* < 0.001	●IVR without SOD: reduction 42% ●IVR with SOD: reduction 70%
30.	Cengiz et al. (2019)	12.856	6.866	*p* = 0.036	
31.	Kao et al. (2019)	Total: 1.9 CV/SICU: 4.5 SICUs: 2.1 MICUs: 0.5	Total: 1.5 CV/SICU: 4.5 SICUs: 1.4 MICUs: 1.0	Total: *p* = 0.005 CV/SICU: *p* = 0.5391 SICUs: *p* < 0.001 MICUs: *p* = 0.0489	
32.	Sousa et al. (2019)	Total: 7.89 ICU A: 4.0% ICU B: 2.4% ICU C: 7.1%	Total: 6.81 ICU A: 4.7% ICU B: 2.1% ICU C: 3.5%	Total: *p* = 0.552 ICU A: *p* = 0.539 ICU B: *p* = 0.001 ICU C: *p* = 0.02	
33.	Branco et al. (2020)	7.99	4.28	*p* < 0.001	
34.	Fortaleza et al. (2020)	36.58	12.04	*p* < 0.001	
35.	Liu et al. (2020)	18.85	13.70	*p* = 0.019	
36.	Michelangelo et al. (2020)	6.11	3.55	*p* < 0.01	
37.	Ochoa et al. (2020)	8.2	3.1	*p* = 0.019	
38.	Shaban et al. (2021)	62.20%	26.90%	*p* < 0.001	

Abbreviations: CV: Cardiovascular; EVAP (onset ≤ 7 days after intubation): early-onset ventilator associated pneumonia; ICU: Intensive Care Unit; IVR: intervention VAP rate; LVAP (onset > 8 days after intubation): late-onset ventilator associated pneumonia; SICU: surgical intensive care unit; VAP: ventilator-associated pneumonia; MICU: Medical Intensive Care Unit; SOD: Selective Oropharyngeal Decontamination. All VAP rates are expressed per 1000 ventilator-days.

## Data Availability

Not applicable.

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
