# Peer review of "The Impact of Care Bundles on Ventilator-Associated Pneumonia (VAP) Prevention in Adult ICUs: A Systematic Review"

_antibiotics, 2023, doi:10.3390/antibiotics12020227_

Round 1

Reviewer 1 Report

The authors report on the systematic review manuscript entitled: The impact of care bundles on the ventilator – associated pneumonia (VAP) prevention in adult ICUs: a systematic review”.

The authors could successfully identify an important gap in this area, but the unnecessarily rigid filtering, very brief review and discussion reduced the quality of the manuscript in addition to some other minor presentation issues such as size of paragraphs and shallow contents.

The following are step by step review:

Line numbering would have made it a bit easier to find comments, but it is not a major issue, I’ll use the section names to review.

Abstract: ….I’d say …a most common, there are many risks in invasive procedures, or

…that have been used to, or simply used to

Fix the word, inclusion, or this might be due to journal processing

…write out on first occurrence Peptic ulcer disease (PUDprophylaxis (we gave 50 mg ranitidine every Deep vein thrombosis (DVT), then use short form afterwards

2. Thirty-six of the included studies demonstrated VAP reduction with a range between 13% and 85%. This is a really large gap showing only the reduction rates; however, no mention of which ones were at the lowest range and which were at the top. Presenting those successful and poor approaches in VAP reduction would give more explanations and good presentation of your results. In fact, this is the only line that presents your result, and it looks odd that so much screening review ends up in a line?

Introduction

Introduction is kind of short, three short paragraphs citing events using only names, rather than revealing tangible initiatives and reviewing major outcomes and reasoning to justify each. After this, then you construct and build a systematic review process deducing outcomes and drawing conclusions to shed light on future gaps for vertical research. This concept needs to be more visible, though it is there in a low profile.

1st Paragrph:…it is a scientifically, instead of scientific

3rd paragraph: …I think you mean, “chronologically”?

Material and Method,

2.1. The review period starting 2005 was a critical period of CA-MRSA and HA-MRSA pandemic that caused significant necrotizing pneumonia frequently accompanied by cytokine storms manifestations that widely required ventilators; not how you’d assess a pandemic situation vs sporadic cases usually endemic in most hospitals.

2.2 …To be included ….revise

3.1 Figure 1. I think you may need to clear this chart for the reader; numbers are not immediately clear unless I do some work to get the numbers you show. As I understand, you started with 3984 minus large number of duplicates (1st round screening) 967 = 3017. But the chart does say that, it says you got 967 after removal of duplicates. Then, records screened is 3984 (not 3017), after 1st round of filtering you got 3017, not sure how 2870 which is the 2nd exclusion result as you show arrows for subsequent flow. I don’t think you need the side boxes showing why you excluded, just excluded for not conforming to criteria would be enough. Instead, please show in the main boxes of eligible hits, which years showed max as I’m more interested in how were the rates around 2005 and after. Those years should be carefully assessed as pandemic periods, you may even mention this outbreak period that preceded other viral outbreaks of SARS-1, MERS, SARS-2. 147 is the 3rd round, not total articles assessed for eligibility, you’re applying criteria sequentially filtering until you ended up with 38 articles eligible…please clarify this

Characteristics of studies: is very brief, it just showed locations, durations, size range…characterize where there more descriptive, cohort, some details will help, what was the characteristics of majority

3.2 the risk of bias (serious and critical should be treated equally) = 11/38 = 30 significant, not sure if you need to include these!

3.3, please join these two paragraphs and elaborate a bit on these cases, only a few lines won’t give a clear understanding of the trends followed at the time and it is difficult to predict what else was done other than the few things you showed, an example is to what degree Head was elevated, which level worked best mostly.

3.4 VAP, please write out after a period. The range 13% to 85% is very wide!, at least mentioned in the text what were those at the higher level and those at the lower level. Stat significant was not mentioned or was not significant, then only the 24 were to consider for positive evaluations.  There is a great deal of variability.

Table 1. Many do not show sample size and age! How inclusion criteria met

The study years were high times for (China, Middle East, USA for MERS, SARS-1, MRSA, and China, USA, Europe and the globe for SARS-2, how come they’re low)

Table 2, title you mean revised?, this table has a good information which is not fully presented in the text, and which be used to synthesize and produce good conclusions. Specific bundles and regions used and combined with significance and reduction rates (Table 3), can help guide a good practice.

Discussion:

Discussion is short. Paragraphs are short, and paragraph leading points are very brief, the trends in different studies and justification for that is not well discussed. Which organisms are likely or are presented in which regions, what were the VAP elements used, which seems to work with which microbes, why? These are important discussions that help reach improvement. Only about 18 refs were discussed, while 56 reviewed.  

Line 16 is already shortened in brackets, just write out without abbreviating again

Line 21, please write out after a period 

Line 53, 54, insert ref numbers, in text citations

The last two paragraphs showed some interesting discussions, but very briefly.

For your limitation, 1st off, as it is a  common pitfall for most systematic review, you applied vigorous filtering and lost some good quality work, 2nd study heterogeneity is not your limitation; however, reaching to such a great heterogeneity comes from your design

Conclusion: you may want to highlight elements that worked in bundles in some regions, those that do not work, justifying reasons, and what were the type of infections in infectious pneumonia requiring VA

Author Response

Dear Reviewer,

Thank you for giving us the opportunity to revise our manuscript entitled "The impact of care bundles on the ventilator – associated pneumonia (VAP) prevention in adult ICUs: a systematic review". We would also like to thank you for your insightful comments and suggestions on how to improve our manuscript. We respectfully tried to address the issues raised and to revise our manuscript accordingly. We hope that our revision will reach the high standards of the journal “Antibiotics”.

We are grateful for your comments. You really help us to improve our manuscript. We apply all your suggestions in our manuscript.

Also, we made changes in the manuscript according to the other Reviewers’ instructions.

We look forward to hearing from you

Best Regards

The authors

Comments from Reviewer

The authors report on the systematic review manuscript entitled: The impact of care bundles on the ventilator – associated pneumonia (VAP) prevention in adult ICUs: a systematic review.

The authors could successfully identify an important gap in this area, but the unnecessarily rigid filtering, very brief review and discussion reduced the quality of the manuscript in addition to some other minor presentation issues such as size of paragraphs and shallow contents.

The following are step by step review:

Line numbering would have made it a bit easier to find comments, but it is not a major issue, I’ll use the section names to review.

Abstract: ….I’d say …a most common, there are many risks in invasive procedures, or …that have been used to, or simply used to

Fix the word, inclusion, or this might be due to journal processing

…write out on first occurrence Peptic ulcer disease (PUD) prophylaxis (we gave 50 mg ranitidine every Deep vein thrombosis (DVT), then use short form afterwards

Answer: Done

Dear Reviewer, thank you for your sharp eye. We fix the errors in Abstract.

  1. Thirty-six of the included studies demonstrated VAP reduction with a range between 13% and 85%. This is a really large gap showing only the reduction rates; however, no mention of which ones were at the lowest range and which were at the top. Presenting those successful and poor approaches in VAP reduction would give more explanations and good presentation of your results. In fact, this is the only line that presents your result, and it looks odd that so much screening review ends up in a line?

Answer: Done

Dear Reviewer, thank you for your comments. After taking into consideration your comments, we revised the Abstract. Please see the Abstract.

Introduction

Introduction is kind of short, three short paragraphs citing events using only names, rather than revealing tangible initiatives and reviewing major outcomes and reasoning to justify each. After this, then you construct and build a systematic review process deducing outcomes and drawing conclusions to shed light on future gaps for vertical research. This concept needs to be more visible, though it is there in a low profile.

Answer: Done

We extended the Introduction taking into consideration all your comments. Please see the Introduction section.

1st Paragrph:…it is a scientifically, instead of scientific

Answer: Done

We fix it.

3rd paragraph: …I think you mean, “chronologically”?

Answer: Done

Yes, you are right.

Material and Method,

2.1. The review period starting 2005 was a critical period of CA-MRSA and HA-MRSA pandemic that caused significant necrotizing pneumonia frequently accompanied by cytokine storms manifestations that widely required ventilators; not how you’d assess a pandemic situation vs sporadic cases usually endemic in most hospitals.

Answer: Done

Dear Reviewer, your comments were taken into account and all the above are mentioned and analyzed in the Introduction, with its present form. Please see the Introduction section.

2.2 …To be included ….revise

Answer: Done

We fix it.

3.1 Figure 1. I think you may need to clear this chart for the reader; numbers are not immediately clear unless I do some work to get the numbers you show. As I understand, you started with 3984 minus large number of duplicates (1st round screening) 967 = 3017. But the chart does say that, it says you got 967 after removal of duplicates. Then, records screened is 3984 (not 3017), after 1st round of filtering you got 3017, not sure how 2870 which is the 2nd exclusion result as you show arrows for subsequent flow. I don’t think you need the side boxes showing why you excluded, just excluded for not conforming to criteria would be enough. Instead, please show in the main boxes of eligible hits, which years showed max as I’m more interested in how were the rates around 2005 and after. Those years should be carefully assessed as pandemic periods, you may even mention this outbreak period that preceded other viral outbreaks of SARS-1, MERS, SARS-2. 147 is the 3rd round, not total articles assessed for eligibility, you’re applying criteria sequentially filtering until you ended up with 38 articles eligible…please clarify this

Answer: Done

Dear Reviewer you are right. We apologize for the mistakes in the Figure 1. Please see the revised Figure 1. Moreover, viral outbreaks, pandemics periods and their possible influence in patients’ outcomes in the revised studies, are all analyzed in the Discussion, with its present form. Please, see the Discussion section.

Characteristics of studies is very brief, it just showed locations, durations, size range…characterize where there more descriptive, cohort, some details will help, what was the characteristics of majority

Answer: Done

Dear Reviewer, please see the section “3.2. Characteristics of the studies”.

3.2 the risk of bias (serious and critical should be treated equally) = 11/38 = 30 significant, not sure if you need to include these!

Answer: Done

We fix it. We treat equally studies with serious and critical bias. Dear Reviewer, please let us include all studies in the review in order to present the literature in total.

3.3, please join these two paragraphs and elaborate a bit on these cases, only a few lines won’t give a clear understanding of the trends followed at the time and it is difficult to predict what else was done other than the few things you showed, an example is to what degree Head was elevated, which level worked best mostly.

Answer: Done

Dear reviewer, we joined 3.4 “Ventilator – Associated Care Bundles” and 3.5 “Patients’ outcomes”, in order to present better the systematic review’s results after your comments. Please, see the section “3.4. Ventilator-Associated Care Bundles – Patients’ outcomes”.

3.4 VAP, please write out after a period. The range 13% to 85% is very wide!, at least mentioned in the text what were those at the higher level and those at the lower level. Stat significant was not mentioned or was not significant, then only the 24 were to consider for positive evaluations.  There is a great deal of variability.

Answer: Done

Dear reviewer, we expanded this section of results as you suggest. Please, see the section “3.4. Ventilator-Associated Care Bundles – Patients’ outcomes”.

Table 1. Many do not show sample size and age! How inclusion criteria met

Answer: Done

Dear Reviewer, we examined again all the revised studies, and we tried to make the table as complete as it can be according to the studies’ data. In some studies, was not mentioned the total number of patients, so we entered the total ventilator days (when this information was available to us). Moreover, when the examined study did not report the mean age, but the median or the interquartile range, this information was also completed in our table. Please, see Table 1.

The study years were high times for (China, Middle East, USA for MERS, SARS-1, MRSA, and China, USA, Europe and the globe for SARS-2, how come they’re low)

Answer: Done

Dear Reviewer, we took your comments into consideration and our analysis is found in Discussion. Please, see Discussion section.

Table 2, title you mean revised?

Answer: Done

You are right. We fix it.

Table 2 has a good information which is not fully presented in the text, and which be used to synthesize and produce good conclusions. Specific bundles and regions used and combined with significance and reduction rates (Table 3), can help guide a good practice.

Answer: Done

Dear Reviewer, thank you for your sharp eye. The full analysis of the Tables 2 and 3 are demonstrated at Results and at Discussion, with their present form. Please, see Results and Discussion sections.

Discussion:

Discussion is short. Paragraphs are short, and paragraph leading points are very brief, the trends in different studies and justification for that is not well discussed. Which organisms are likely or are presented in which regions, what were the VAP elements used, which seems to work with which microbes, why? These are important discussions that help reach improvement. Only about 18 refs were discussed, while 56 reviewed.  

Answer: Done

Dear Reviewer, we expand the Discussion according to your comments. Please see Discussion section.

Line 16 is already shortened in brackets, just write out without abbreviating again

Answer: Done

We fix it.

Line 21, please write out after a period 

Answer: Done

We fix it.

Line 53, 54, insert ref numbers, in text citations

Answer: Done

The last two paragraphs showed some interesting discussions, but very briefly.

Answer: Done

Dear Reviewer, we expand the Discussion according to your comments. Please see Discussion section.

For your limitation, 1st off, as it is a  common pitfall for most systematic review, you applied vigorous filtering and lost some good quality work, 2nd study heterogeneity is not your limitation; however, reaching to such a great heterogeneity comes from your design

Answer: Done

Dear Reviewer, we take into consideration your comments. Please, see the Limitations section.

Conclusion: you may want to highlight elements that worked in bundles in some regions, those that do not work, justifying reasons, and what were the type of infections in infectious pneumonia requiring VA

Answer: Done

Dear Reviewer, we revised Conclusions according to your comments. Please, see Conclusions section.

Reviewer 2 Report

It is a complehensive and well design study of metaanalysis and sysematic review of VAP care bundles. It is a interesting and worthy to read for clinicians and ICU health care workers. 

Author Response

Thank you very much for your comments regarding the manuscpript

Reviewer 3 Report

This paper entitled "The impact of care bundles on the ventilator – associated pneumonia (VAP) prevention in adult ICUs" describing the importance and improvement of using ventilator in ICU  which is one of the most common pneumonia infection causing agent for the patient. However it needs a lot of improvements: First, rearrange the references and start the first reference with number one and so on....... result section have not adequately written, please explain the merit and demerit of each latest version of guideline for using of ventilator in ICU.  Even after after following of international standard guideline why some country have more infection and other have less ? is it due to non trained medical stop or poor healthcare system. Provide the most common infectius stain name along with country.

Author Response

Dear Reviewer,
Thank you for giving us the opportunity to revise our manuscript entitled "The impact of care bundles on the ventilator - associated pneumonia (VAP) prevention in adult ICUs: a systematic review". We would also like to thank you for your insightful comments and suggestions on how to improve our manuscript. We respectfully tried to address the issues raised and to revise our manuscript accordingly. We hope that our revision will reach the high standards of the journal "Antibiotics".
We are grateful for your comments. You really help us to improve our manuscript. We apply all your suggestions in our manuscript.
Also, we made changes in the manuscript according to the other Reviewers' instructions.
We look forward to hearing from you
Best Regards
The authors

Comments from Reviewer
This paper entitled "The impact of care bundles on the ventilator - associated pneumonia (VAP) prevention in adult ICUs" describing the importance and improvement of using ventilator in ICU which is one of the most common pneumonia infections causing agent for the patient.
However it needs a lot of improvements:
First, rearrange the references and start the first reference with number one and so on.
Answer: Done
Dear Reviewer, thank you for your sharp eye. We rearranged the references and we started the first reference with number one and so on.

Result section have not adequately written, please explain the merit and demerit of each latest version of guideline for using of ventilator in ICU. Even after following of international standard guideline why some country have more infection and other have less? is it due to non trained medical stop or poor healthcare system. Provide the most common infectious stain name along with country.
Answer: Done
Dear Reviewer, thank you for your very useful comments. After taking into consideration your comments, we revised and expanded the Results and Discussion sessions. Please see the Results and Discussion sessions.

Round 2

Reviewer 1 Report

The manuscript is improved now. thank you for taking the time to do that. However, I still have minor issues. 

section 3.3 my suggestion was to join the tiny paragraphs, but I think you re-wrote and rephrased, its ok.

L 122, write out

Ll125, write out

In limitations and conclusions; re-visit how you presented the "heterogeneity" and "non-heterogeneity"

My concern is that throughout the manuscript, paragraphs are tiny, often with about four lines.

References look good

I think the authors did a great job ion improving, I'm satisfied.

Author Response

Dear Reviewer,

Thank you for giving us the opportunity to revise our manuscript entitled "The impact of care bundles on the ventilator – associated pneumonia (VAP) prevention in adult ICUs: a systematic review". We would also like to thank you for your insightful comments and suggestions on how to improve our manuscript. We respectfully tried to address the issues raised and to revise our manuscript accordingly. We hope that our revision will reach the high standards of the journal “Antibiotics”.

We are grateful for your comments. You really help us to improve our manuscript. We apply all your suggestions in our manuscript.

We look forward to hearing from you

Best Regards

The authors

The manuscript is improved now. thank you for taking the time to do that. However, I still have minor issues. 

section 3.3 my suggestion was to join the tiny paragraphs, but I think you re-wrote and rephrased, its ok.

Answer: Done

Dear Author, you are right. We re-wrote this section

L 122, write out

Ll125, write out

Answer: Done

We write out the Lines 122 and 125.

In limitations and conclusions; re-visit how you presented the "heterogeneity" and "non-heterogeneity"

Answer: Done

Dear Author, you are right and thank you especially for this comment. We misused the term "non-heterogeneity" in the conclusions. Thus, we remove the term "non-heterogeneity" from the conclusions in order to avoid confusion. Also, we further explain "non-heterogeneity" in the Limitations section. Please, see the Limitations and Conclusions sections.

My concern is that throughout the manuscript, paragraphs are tiny, often with about four lines.

Answer: Done

Dear Author, we re-arrange some paragraphs and now two paragraphs are about four lines.

References look good

I think the authors did a great job on improving, I'm satisfied.

Reviewer 3 Report

The Authors have answered all my criticisms very well, and i am satisfied with them.

Author Response

Thank you very much for your comments